# Early planetesimal differentiation and late accretion shaped Earth's nitrogen budget

Wenzhong Wang [1,2,3,4] ✉, Michael J. Walter [3], John P. Brodholt [4,5] & Shichun Huang [6]

The relative roles of protoplanetary differentiation versus late accretion in establishing Earth's life-essential volatile element inventory are being hotly debated. To address this issue, we employ first-principles calculations to investigate nitrogen (N) isotope fractionation during Earth's accretion and differentiation. We find that segregation of an iron core would enrich heavy N isotopes in the residual silicate, while evaporation within a $H_2$-dominated nebular gas produces an enrichment of light N isotope in the planetesimals. The combined effect of early planetesimal evaporation followed by core formation enriches the bulk silicate Earth in light N isotopes. If Earth is comprised primarily of enstatite-chondrite-like material, as indicated by other isotope systems, then late accretion of carbonaceous-chondrite-like material must contribute ~30–100% of the N budget in present-day bulk silicate Earth. However, mass balance using N isotope constraints shows that the late veneer contributes only a limited amount of other volatile elements (e.g., H, S, and C) to Earth.

How terrestrial planets accreted their life-essential volatile elements, such as H, N, C, and S, is a matter of continued debate[1]. One of the most popular models is the late veneer hypothesis[2–4], under which the proto-Earth accreted from nearly volatile-free material[5,6], possibly as a consequence of volatile loss due to the extensive heating and melting caused by impact accretion and the decay of short-lived nuclei such as $^{26}$Al in the earliest-formed planetesimals[7]. If Earth accreted from volatile-poor materials, then its volatile elements may have been primarily added by a late veneer of chondrite-like material likely originating in the volatile-rich, outer reaches of the solar system[6]. Alternatively, Earth might have accreted from volatile-rich materials, with the current volatile element abundances set during Earth's main growth stage as a consequence of evaporative loss[4,8,9] and/or partitioning of elements into the Earth's core[3], and a late veneer is not needed. Distinguishing between these two possible mechanisms is crucial for understanding how

volatile elements were delivered to Earth and other terrestrial planets.

Earth's nitrogen isotopic composition can provide constraints on the origins of Earth's volatile elements. Based on the estimated abundance and isotopic composition of N ($\delta^{15}N = [(^{15}N/^{14}N)_{sample}/(^{15}N/^{14}N)_{atm} - 1] \times 1000‰$, where $(^{15}N/^{14}N)_{atm}$ refers to $^{15}N/^{14}N$ in the atmosphere[10]) in the atmosphere ($\delta^{15}N_{atm} = 0‰$), crust ($\delta^{15}N_{crust} \approx +6‰$), and mantle ($\delta^{15}N_{mantle} = -5 \pm 4‰$)[11,12], the N isotopic composition of the bulk silicate Earth ($\delta^{15}N_{BSE}$) is estimated to be $-1.5 \pm 3‰$[13]. This value is distinct from the $\delta^{15}N$ of enstatite chondrites ($\delta^{15}N \sim -47$ to $-10‰$)[14,15] (see Supplementary Materials), the primitive meteorite group thought to be most representative of Earth's building block[16,17], and most carbonaceous chondrites ($\delta^{15}N \sim +10$ to $+56‰$)[18,19]. If Earth accreted nearly volatile-free, then a late veneer mixture of enstatite and carbonaceous chondrites might reproduce the present-day N abundance and isotopic signature of the bulk silicate Earth. However, constraining the nature of

[1]Deep Space Exploration Lab/School of Earth and Space Sciences, University of Science and Technology of China, Hefei, Anhui 230026, China. [2]CAS Center for Excellence in Comparative Planetology, University of Science and Technology of China, Hefei, Anhui, China. [3]Earth and Planets Laboratory, Carnegie Institution for Science, Washington, DC 20015, USA. [4]Department of Earth Sciences, University College London, London, WC1E 6BT, UK. [5]The Centre of Planetary Habitability, University of Oslo, Oslo, Norway. [6]Department of Earth, Environmenral, & Planetary Sciences, University of Tennessee at Knoxville, Knoxville, TN, USA. ✉e-mail: wwz@ustc.edu.cn

the late veneer based on N isotopes requires knowledge of how protoplanetary differentiation processes, such as core formation and evaporation, fractionate N isotopes among different reservoirs. Unfortunately, these essential data are currently insufficiently understood.

Many experimental studies show that N behaves as a siderophile element and would have been largely sequestered into the metallic core[3,20–26]. In contrast, experimental investigations of the N isotope fractionation between metal and silicate[21,22,26] performed at pressures <7 GPa, much lower than those under which Earth's core formed[27], show large discrepancies in their measured isotopic fractionation factors[21,22,26,28], precluding any definitive conclusion on the effect of core formation. Moreover, previous studies using first-principles calculations[26,29] also reported the equilibrium N isotope fractionation factors among N-bearing minerals and molecules at 0 GPa, but these inter-mineral fractionation factors cannot be used to model the N isotope fractionation between silicate and metallic melts due to the large difference in structures between melts and crystals. There is only one study in the literature that has investigated the N isotope fractionation during evaporative degassing from a magma ocean[30]. Consequently, it is not possible to use the literature N isotope fractionation data to robustly understand the origins of the non-chondritic $\delta^{15}N_{BSE}$.

In this work, we conducted first-principles calculations (see details in Supplementary Materials) to constrain the equilibrium N isotope fractionation factors ($10^3 \ln\alpha$) for both silicate-metal and vapor-silicate. Using our results, we constrained the N isotope fractionation during protoplanetary differentiation and further investigated the origin of Earth's volatile elements.

## Results

### Structural properties and force constants of nitrogen in melts

We conducted first-principles molecular dynamics (FPMD) simulations to obtain the structures of N-bearing silicate and metallic melts at 0–99 GPa and 3000 K. Experiments[22,23,31–33] show that N dissolves principally as $N_2$ in silicate melts under relatively oxidizing conditions (log $fO_2$ > IW-1.5; oxygen fugacity reported in log units relative to the iron-wüstite buffer, IW) but as $N^{3-}$ under relatively reducing conditions in the form of N-Si and/or N-H (log $fO_2$ < IW-1.5). Therefore, we consider two kinds of silicate melts representing oxidizing and reducing conditions, respectively. We model the relatively oxidizing conditions using $Mg_{32}Si_{32}O_{95}N_2$ and $Mg_{30}NaCa_2Fe_4Si_{24}Al_3O_{89}N_2$ ("pyrolite+$N_2$") melt compositions. In these cases, N occurs dominantly as $N_2$ with a short N-N bond length of 1.11–1.15 Å (Supplementary Figs. 1 and 3). For relative reduced systems we simulate $Mg_{32}Si_{32}O_{96}NH_3$ and $Mg_{30}NaCa_2Fe_4Si_{24}Al_3O_{89}NH_3$ ("pyrolite+$NH_3$") melts in which N is mainly bonded to Si with an N-Si distance of ~1.72 Å (Supplementary Figs. 2 and 3). In addition, there are also significant amounts of N-H bonds in $Mg_{32}Si_{32}O_{96}NH_3$ melt and N-Fe and N-Mg bonds in $Mg_{30}NaCa_2Fe_4Si_{24}Al_3O_{89}NH_3$ melt. Our results are consistent with the experimental findings of N species in silicate melts under different $fO_2$ conditions[22,23,31–33]. In $Fe_{98}N_2$ and $Fe_{87}Ni_4Si_6S_2C_2O_5H_5N_2$ metallic melts, N is dominantly bonded to Fe atoms with an N-Fe distance of ~1.84 Å. The presence of other light elements does not significantly change the N bonding environment (Supplementary Fig. 4).

The force constant <F> of N in silicate and metallic melts (Supplementary Table 1) is controlled by melt structure. Generally, a shorter bond has a stronger bond strength and a larger <F>[34–36]. Therefore, <F> increases in the order of N-Fe in metal <$N^{3-}$ in silicate <$N_2$ in silicate (Fig. S9). Because the N-N bond length of $N_2$ in the $Mg_{32}Si_{32}O_{95}N_2$ melt does not significantly change with pressure (Supplementary Fig. 1), its <F> does not substantially change with pressure either, with a value of ~760 N/m within the pressure range explored in our study. In contrast, the <F> of N in $Mg_{32}Si_{32}O_{96}NH_3$ and $Fe_{98}N_2$ melts increases by ~70% from 0 to 99 GPa, mainly reflecting an increase in the coordination numbers (CNs) of N-Si and N-Fe with pressure

(Supplementary Figs. 2 and 4). A comparison between $MgSiO_3$ and pyrolitic melts shows that other components have a limited effect on the <F> of N in silicate melts. In metallic melts, $Fe_{98}N_2$ and $Fe_{87}Ni_4Si_6S_2C_2O_5H_5N_2$ have similar <F> values, consistent with their similar N-Fe bonding.

### Core-mantle nitrogen isotope fractionation

The $10^3 \ln\alpha$ between silicate and metallic melts ($10^3 \ln\alpha_{silicate-metal}$) is derived from the differences in <F> using the high-temperature approximation of the Bigeleisen-Mayer equation[37]. Silicate melts are always enriched in $^{15}N$ relative to metallic melts, but the degree of enrichment is affected by the N species in silicate melts as well as pressure and temperature (Fig. 1). At 3000 K, the $10^3 \ln\alpha$ between the oxidizing $Mg_{32}Si_{32}O_{95}N_2$ silicate melt ($N_2$ species) and $Fe_{98}N_2$ decreases from ~1.4‰ at 0 GPa to ~1.2‰ at 90 GPa (Fig. 1a). The $10^3 \ln\alpha$ between the reducing $Mg_{32}Si_{32}O_{95}NH_3$ silicate melt ($N^{3-}$ species) and $Fe_{98}N_2$ is smaller than that under relatively oxidizing conditions and increases with pressure (-0.4‰ at 0 GPa and -0.7‰ at 90 GPa). Our results are consistent with the experimental results (1.1–5.5‰) obtained by ref. 21 at 1.5–7 GPa and 1873–2073 K, although their results have large uncertainties (>3‰) and show no pressure or temperature dependence.

Reference[22] reported the N isotope fractionation factor between silicate and metal ($\Delta^{15}N_{silicate-metal}$) ranging from 49 to 257‰ at -IW-3 to IW-0.5, at 1 GPa, 1673 K, one or two orders of magnitude greater than that obtained in our calculations. If these reported high values represent equilibrium fractionation factors, then the inferred <F> difference between silicate and metal must be unrealistically high, 6400–33800 N/m, which is 10–50 times the <F> of N in $N_2$. Reference[28] suggested that $\Delta^{15}N_{silicate-metal}$ ranges from −10‰ at IW-5 to +5‰ at IW, at ~2000 K, corresponding to a <F> difference of −1910 N/m to +955 N/m between silicate and metal if equilibrium. However, such required values also cannot be realistic as their magnitudes are much larger than that of the N-N bond in $N_2$, the species which has the maximum <F>. More recently, Grewal et al.[26] suggested that $\Delta^{15}N_{silicate-metal}$ increases from +1.0‰ to +3.3‰ at IW-3.8 to IW-1.7, at 2–3 GPa and 1673–2073 K, much smaller than previous experimental results[22,28]. The newly reported $\Delta^{15}N_{silicate-metal}$ and its dependence on $fO_2$ are consistent with our calculations at low pressures (Fig. 1).

Using our results, we model N isotope fractionation under the conditions of Earth's core formation[27]. Following a Rayleigh distillation model, our results show that when considering the entire range of experimentally determined metal-silicate partition coefficients[3], core-mantle differentiation can only shift the $\delta^{15}N_{BSE}$ by at most +2‰ and +5‰ under relatively reducing and oxidizing conditions, respectively. The magnitude of fractionation becomes much smaller (<+1.2‰) when using an equilibrium core-formation process (Supplementary Fig. 10). Our results show that if Earth accreted from an enstatite chondrite-rich mixture or from most carbonaceous chondrite materials, core formation cannot explain the observed $\delta^{15}N_{BSE}$.

### Nitrogen isotope fractionation caused by planetesimal evaporation

We now consider the N isotope effect during evaporative loss from molten planetesimals caused by heat from impact accretion and the decay of short-lived nuclei such as $^{26}Al$. The net isotope fractionation between vapor and melt ($\Delta^{15}N_{vapor-melt}$) could be equilibrium or kinetic fractionation, depending on the evaporation conditions[38]. If evaporation is dominated by the kinetic effect, the melt would always become enriched in heavy isotopes after evaporation, which cannot explain the sub-chondritic S isotope composition of the bulk silicate Earth[9]. Equilibrium isotope fractionation during planetesimal evaporation can explain the Mg, Si, Se, and Te isotopic and elemental compositions of bulk Earth[38,39]. This may correspond to the case that planetesimals undergo evaporation in the presence of nebular $H_2$ gas with a

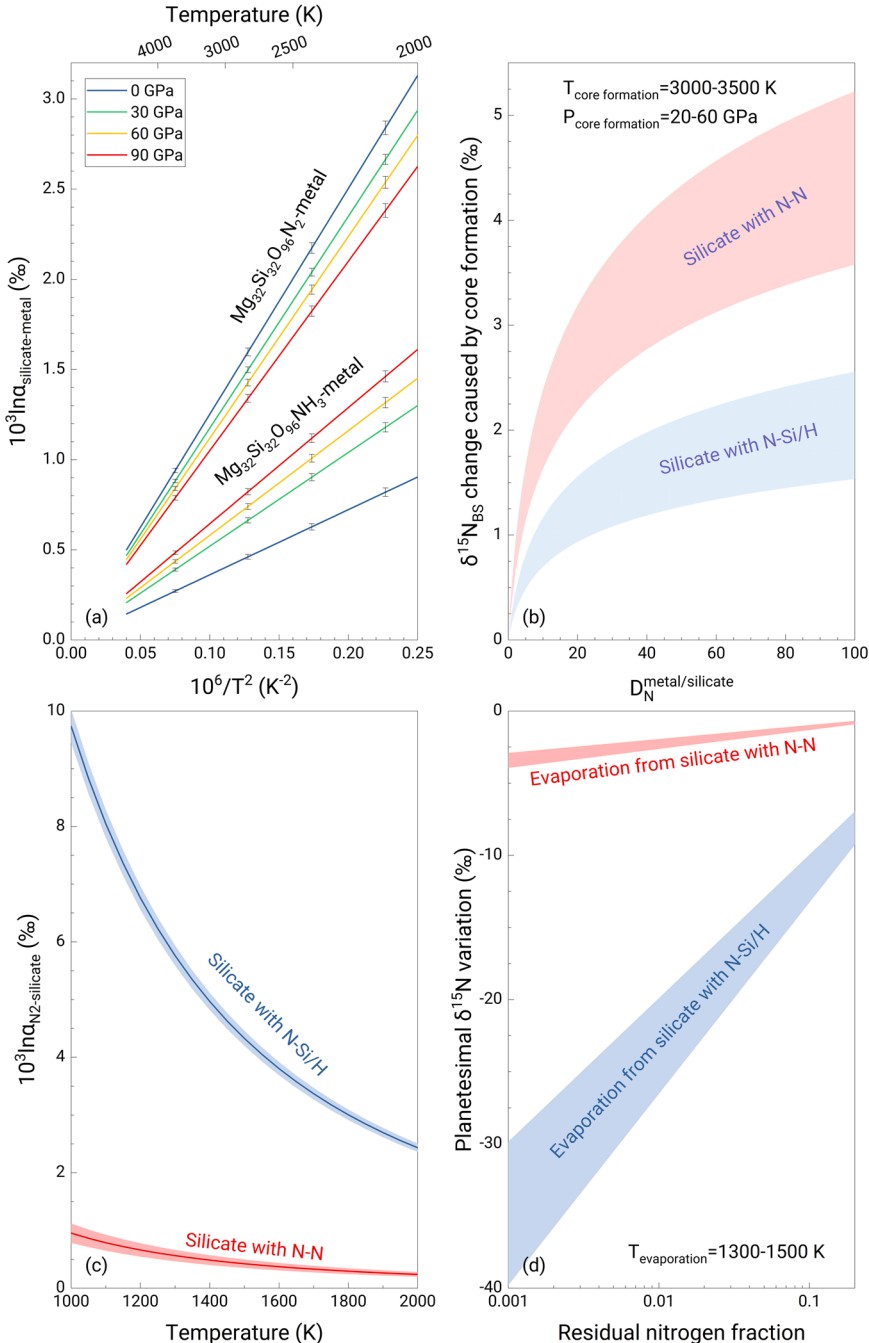

**Fig. 1 | Nitrogen isotope fractionation during core formation and planetesimal evaporation. a** The equilibrium N isotope fractionation between silicate and metallic melts ($10^3\ln\alpha_{silicate-metal}$) under relatively oxidizing and reducing conditions as a function of temperature at different pressures. Error bars represent the $\pm 1\sigma$ deviation derived from propagating the $\pm 1\sigma$ deviation of the force constant. **b** The $\delta^{15}N$ change in the bulk silicate reservoir (mantle + crust + atmosphere) caused by core-mantle differentiation under the conditions of Earth's core formation[27] following a Rayleigh distillation model. Core-forming pressure and temperature are

20–60 GPa and 3000–3500 K, respectively. $D^{metal/silicate}_N$ is the N partition coefficient between metal and silicate. The maximum N isotopic effect caused by core formation is less than +5‰. **c** the equilibrium N isotope fractionation between vapor (dominantly occurring as $N_2$) and silicate melt ($10^3\ln\alpha_{N2-silicate}$) under relatively reducing and oxidizing conditions. The temperature for planetesimal evaporation is set to be 1300–1500 K. **d** the $\delta^{15}N$ change of planetesimals caused by evaporation as a function of residual N fraction.

protostellar pressure of approximately $10^{-4}$ bar. As such, $\Delta^{15}N_{vapor-melt}$ during evaporation is equal to the equilibrium isotope fractionation between vapor and melt ($10^3\ln\alpha_{vapor-silicate}$). Based on that framework, we conducted thermodynamic calculations using solar abundances for the elements[40] to determine the N species in the vapor phase. The results show that $N_2$ is always the dominant vapor species (>99.9%) regardless of the H concentrations in the system (Supplementary Fig. 11), consistent with experimental observations[30]. Therefore, the

$10^3\ln\alpha_{vapor-silicate}$ is equal to $10^3\ln\alpha_{N2-silicate}$. At 1300–1500 K, the $10^3\ln\alpha_{vapor-silicate}$ is +4.3-5.8‰ and +0.4-0.6‰ (Fig. 1c) under relatively reducing and oxidizing conditions, respectively. Evaporative loss of 99% of the accreted N causes a negative shift of −26 to −20‰ in the $\delta^{15}N$ of a planetesimal under relatively reducing conditions, while this shift is only ~−2.5‰ under relatively oxidizing conditions (Fig. 1d).

These fractionation values are much lower than those in ref. 30. They reported a decrease of up to −41 ± 13‰ in $\delta^{15}N$ of the melt when

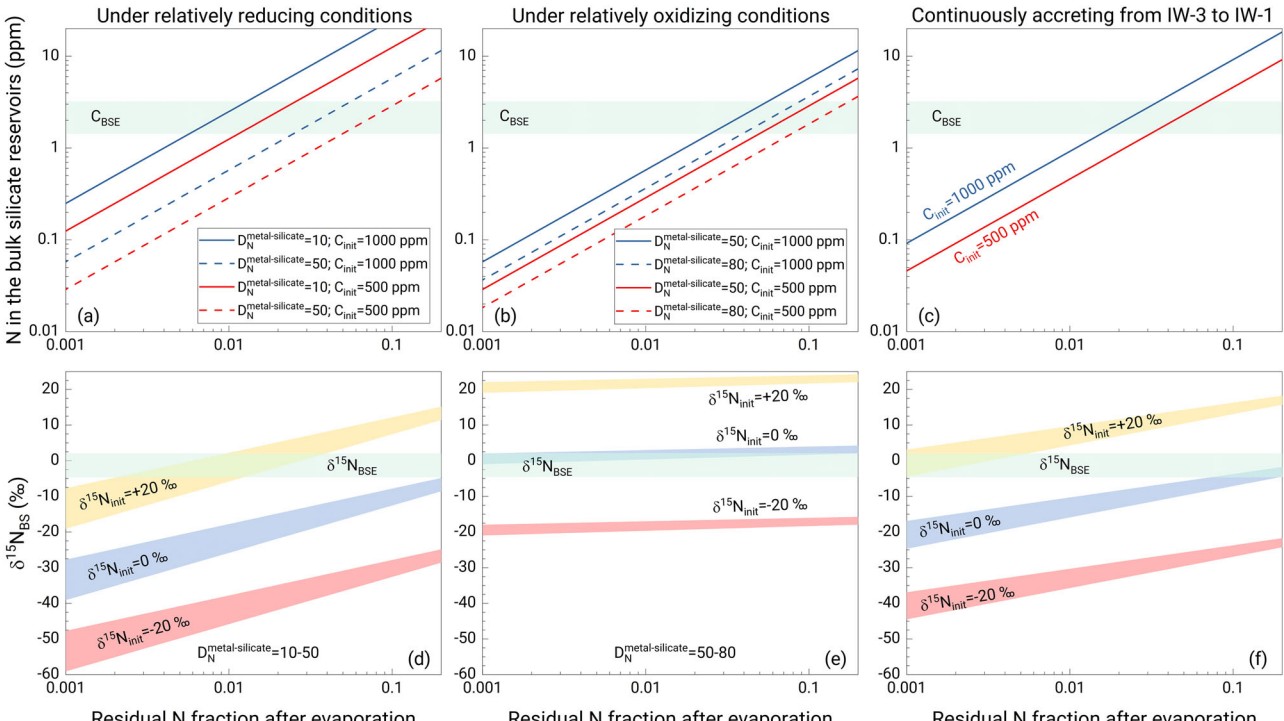

**Fig. 2 | Combined effect of planetesimal evaporation and core formation on the N abundance and N isotope composition of rocky planets. a–c** The N abundance and (**d–f**) $\delta^{15}N$ in the bulk silicate reservoir as a function of the residual N fraction after evaporation. (**a, d**) Under relatively reducing conditions; (**b, e**) Relatively oxidizing conditions. (**c, f**) The modeled N concentration and $\delta^{15}N$ in the bulk silicate part at the oxygen fugacity (log $fO_2$) of Earth's accreting materials (-IW-3 to IW-1). Earth's core/mantle mass ratio was also used for models in (**a**) and (**b**). The $fO_2$ affects the N partition coefficient between metal and silicate (D$^{metal/silicate}_N$) and the N species in silicate melts[21,22,24,31,33], which consequently controls the equilibrium N isotope fractionation between vapor (N$_2$) and silicate melt ($10^3$ln$\alpha_{N2\text{-silicate}}$) (Fig. 1c). The green areas represent the values of the bulk silicate Earth. The red and blue lines in upper panels represent the initial N concentrations (C$_{init}$) of 500 and 1000 ppm in planetesimals before evaporation, respectively. The dash lines refer to the modeling results at different D$^{metal/silicate}_N$ values. The yellow, blue, and red shadow regions in the lower panels represent the modeled $\delta^{15}N$ in the bulk silicate reservoir with an initial $\delta^{15}N$ ($\delta^{15}N_{init}$) of +20‰, 0‰, and −20‰, respectively.

53% of the N was degassed at ~IW-2, and an apparent gas-melt N isotope fractionation of ~ +35‰ is required to explain the experimental results[30]. This positive fractionation factor indicates equilibrium fractionation between gas and melt because kinetic fractionation via evaporation enriches the residual melt in heavy isotopes[38]. The authors noted the large fractionation and proposed N diffusion in the melt as a possible explanation[30]. However, such a mechanism would result in enrichment of $^{15}N$ in the melt, not a decrease, because $^{14}N$ diffuses faster than $^{15}N$ and more $^{14}N$ would be lost during degassing.

## Discussion

Combining our isotope fractionation data with the literature N metal-silicate partition coefficients (D$^{metal/silicate}_N$)[3], we model the N abundance (C$_{BS}$) and $\delta^{15}N$ in the bulk silicate reservoir ($\delta^{15}N_{BS}$) for early planetesimal evaporative loss followed by late-stage core formation under a range of conditions (Fig. 2). Because both the N isotope fractionation and D$^{metal/silicate}_N$ are affected by $fO_2$, the modeled C$_{BS}$ and $\delta^{15}N_{BS}$ depend on the $fO_2$ of the accreting materials. We consider models with $\delta^{15}N$ of the accreting materials ranging from −20‰ to +20‰, effectively simulating Earth's accretion from enstatite-chondrite-like to carbonaceous-chondrite-like materials. We assume an initial concentration of N of 500–1000 ppm, approximating the range of enstatite and carbonaceous chondrites[41].

Our models show that the N abundance in the present-day bulk silicate Earth (C$_{BSE}$) can be reproduced by ~90–99% early evaporative loss followed by late-stage core formation over the modeled $fO_2$ range, with slightly less evaporation required under relatively oxidizing conditions (Fig. 2). However, because $\delta^{15}N$ becomes progressively lower with evaporative loss, a planet originating from material with a strongly negative $\delta^{15}N$, such as enstatite-rich material[17], can never evolve to the bulk silicate Earth's $\delta^{15}N$ by combining evaporative loss and core formation, regardless of the $fO_2$. An Earth built from carbonaceous-chondrite-like material with an initial positive $\delta^{15}N$ can evolve to the C$_{BSE}$ and $\delta^{15}N_{BSE}$ only under relatively reducing conditions (Fig. 2a). A starting material with a $\delta^{15}N$ of 0‰, e.g., a mixture of enstatite and carbonaceous chondrites, can also reproduce both C$_{BSE}$ and $\delta^{15}N_{BSE}$, but only under fully oxidizing conditions (Fig. 2b). If the Earth accreted with a uniformly evolving oxygen fugacity from -IW-3 to -IW-1 (refs. 42,43), an initial $\delta^{15}N$ of approximately +10‰ provides a successful solution (Fig. 2c and Supplementary Fig. 13).

Our results show that early evaporation from planetesimals followed by core formation starting with a carbonaceous-chondrite-like composition can explain the $\delta^{15}N_{BSE}$ and C$_{BSE}$. However, data from multiple isotopic systems[17] support that Earth mainly accreted from enstatite-chondrite-like materials with a $\delta^{15}N$ of −47 to −10‰ (ref. 14), and subsequent protoplanetary differentiation would result in an even more negative $\delta^{15}N$ for the bulk silicate Earth. In this case, a late veneer with a positive $\delta^{15}N$ must have contributed to the present-day bulk silicate Earth's N budget to reproduce the present-day $\delta^{15}N_{BSE}$. As most carbonaceous chondrites have positive $\delta^{15}N$ of +10 to +56‰ (ref. 18,19), a late veneer of carbonaceous-chondrite-like material satisfies this criterion, consistent with previous constraints[6]. The amount of N added by the late veneer to the BSE depends on how much N is lost during planetesimal evaporation as well as the isotopic composition of the late veneer material. The more N that remains after accretion, the higher the $\delta^{15}N$ of late-veneer material is needed to move the bulk silicate Earth from an initially negative $\delta^{15}N$ composition to the present-day value and still match the amount of N currently in the bulk

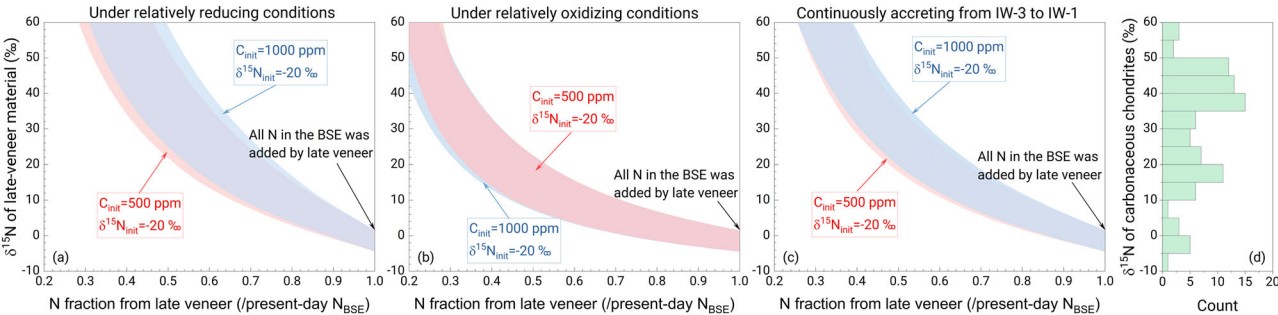

**Fig. 3 | Reproducing the $\delta^{15}N$ in the bulk silicate Earth using planetary processes (evaporation + core formation) and a late veneer.** The $\delta^{15}N$ of late-accretion material versus the N fraction from a late veneer under (**a**) relatively reducing conditions ($D^{metal/silicate}_N = 10-50$); **b** relatively oxidizing conditions ($D^{metal/silicate}_N = 50-80$); and (**c**) at the oxygen fugacity (log $fO_2$) of ~IW-3 to IW-1 for Earth's accreting materials (using the dependence of $D^{metal/silicate}_N$ on log $fO_2$, ref. [3]). The initial $\delta^{15}N$ of Earth's building material ($\delta^{15}N_{init}$) is estimated to be −20‰ according to the best-fit model for the Earth (71% enstatite chondrite + 24% ordinary chondrite + 5% CV/CO chondrite)[17]. Shown in (**d**) is the $\delta^{15}N$ distribution of carbonaceous chondrites (CI and CM)[14,18,19]. CV and CO chondrites data are not included because they have very negative $\delta^{15}N$ values and are not candidates for late veneer

material. The less N remains in the bulk silicate Earth after evaporation and core formation, the more N is added by a late veneer to reproduce the $C_{BSE}$. Because the effect of core formation on the N concentration of the bulk silicate part is well-known, the residual N fraction after evaporation versus the N fraction from a late veneer is determined for different initial N concentrations to match the $C_{BSE}$ (Supplementary Fig. 3). Thus, the $\delta^{15}N$ of the bulk silicate part after protoplanetary differentiation can be derived from the results of Fig. 2. The late veneer material is expected to have a positive $\delta^{15}N$ that is similar or close to those of carbonaceous chondrites. If all N is lost during evaporation, all N in the bulk silicate Earth would be from a late veneer with a $\delta^{15}N$ identical to the $\delta^{15}N_{BSE}$.

silicate Earth (Supplementary Fig. 14). This is shown in Fig. 3 for the same range of conditions as considered in Fig. 2. The multistage accretion model shows that the N abundance in the bulk silicate Earth can be fully established by a late veneer of carbonaceous chondrite contributing between 100% (in which case the $\delta^{15}N$ of late-accreting material is similar to the present-day $\delta^{15}N_{BSE}$) and ~30% of the current N in the bulk silicate Earth. However, if evaporation and core formation were less efficient at removing N and the bulk silicate Earth kept 70% of its original N, then no chondrite composition can explain the present-day $\delta^{15}N_{BSE}$ through late addition.

Our results suggest that both protoplanetary differentiation and a late veneer together control the present-day bulk silicate Earth's N budget for an enstatite-chondrite-like Earth model, of which somewhere between 30% and 100% was contributed by late addition of carbonaceous-chondrite-like material (Fig. 3). Using the abundance of N in carbonaceous chondrites, the mass of the late-accreting material is constrained to be just 0.04–0.2% of the mass of Earth's mantle (Supplementary Fig. 15). This is a relatively small amount and has important consequences for the origins of other elements, in particular, other volatile elements and highly siderophile elements (HSEs) (Fig. 4).

First, a late veneer can only supply a small amount of other volatile elements. For instance, even if the late veneer contributed 100% of the current N in the bulk silicate Earth, it would only supply <5% of the bulk silicate Earth's H abundance[44,45], corresponding to 5–10% of Earth's ocean mass. This indicates that a late veneer cannot establish the H abundance in the BSE and Earth should have accreted its water from its major source material—enstatite-chondrite-like material[46]—and/or through the interaction between primordial hydrogen-rich atmosphere and proto-Earth[47] during early accretion and differentiation. A late veneer can also contribute only 7–45% of the bulk silicate Earth's C abundance[12], suggesting a substantial amount of C in the bulk silicate Earth subsequent to Earth's core formation. This aligns with recent high-pressure experiments, which indicate that under the conditions of core formation for Earth C becomes substantially less siderophile compared to its behavior at lower pressures[48]. Similarly, the late veneer could only supply at most 30% of the present-day bulk silicate Earth's S, Se, and Te budgets (Fig. 4). This is consistent with our previous study showing that the sub-chondritic S isotope signature in the bulk silicate Earth can be achieved mainly through planetesimal evaporation, with no more than ~30% of the present-day bulk silicate Earth's S budget added by a late veneer[9].

Second, the limited mass of a late veneer suggests that at most 30% of the highly siderophile element (HSE) budgets in the bulk silicate Earth were added by a late veneer. This contradicts the widely held view that core formation is likely to remove most, if not all, HSEs from the mantle and that most HSEs in the bulk silicate Earth come from late accretion of chondritic material[49]. However, high P-T experiments yield low metal-silicate partition coefficients for palladium (Pd)[50] and platinum (Pt)[51] and suggest possible high concentrations of platinum-group elements in the mantle after core-mantle differentiation and in turn less contribution from late veneer, consistent with our conclusions based on N. Whether other HSEs would become less much siderophile under high pressures remains an open question. Further investigations into the metal-silicate partition coefficients of these HSEs will help to verify our constraints on the contribution of a late veneer to the HSE abundances in the bulk silicate Earth.

Finally, the small mass of the late veneer estimated using N isotopes is also consistent with a recent estimate from triple oxygen isotopes[52] but is smaller than the estimate based on Ru isotopes[53]. The difference may be reconciled by the late addition of carbonaceous group iron meteorite-like materials (such as IID and IVA irons) that have Ru isotope compositions comparable to those of carbonaceous chondrites but are deficient in oxygen and volatiles[52,54].

Additionally, noble gases provide significant insights into the origin of Earth's volatiles, albeit presenting a more intricate narrative of their origin and evolution[55–59]. For instance, the high Ne isotope ratio observed in the primordial plume mantle indicates the preservation of nebular gases in the deep mantle[59], while the isotopic composition of heavy Kr and Xe, primarily residing in the atmosphere, has been attributed to a late delivery of carbonaceous-chondrite-like material[60] or cometary ice[61]. Further high-precision investigation into mantle reservoirs of noble gases, as well as noble gas isotope fractionation during planetary differentiation[62], holds promise in understanding the origin of Earth's noble gases.

In summary, our results show that protoplanetary differentiation processes, especially planetesimal evaporation, can significantly fractionate N isotopes, but they cannot simultaneously reproduce the N budget and isotopic signature in the bulk silicate Earth. A late accretion of carbonaceous-chondrite-like material providing ~30-100% of the present-day bulk silicate Earth's N budget is required to explain the Earth's nitrogen budget and isotope compositions if Earth accreted

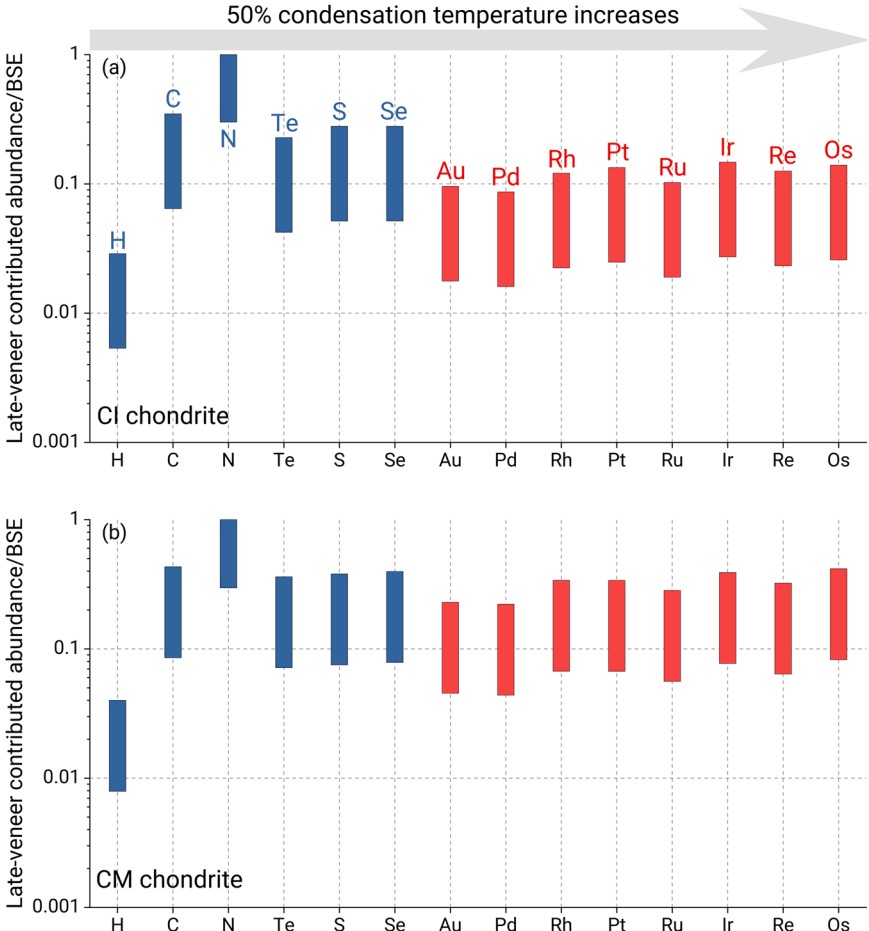

**Fig. 4 | Effect of late veneer on the elemental abundances in the bulk silicate Earth.** The upper (**a**) and lower (**b**) panels represent the late accretion of CI-chondrite-like and CM-chondrite-like materials, respectively. The elemental abundances in the CI and CM chondrites and the BSE are from literature studies[6,12,49,70–74].

The late veneer, which contributed 30–100% of the N in the present-day BSE, is 0.04–0.2% of Earth's mantle by mass, and so, it has a limited effect on the budgets of other volatile elements in the BSE.

from enstatite chondrite-rich material. However, this has a limited effect on the inventory of other volatile elements and HSEs (Fig. 4).

## Methods

### Equilibrium isotope fractionation factor

Equilibrium mass-dependent isotope fractionation arises from the change in vibrational energy caused by isotopic substitution. Following the Bigeleisen–Mayer equation[37], the reduced partition function ratio (β) of isotopes in a phase of interest, which represents the equilibrium isotope fractionation factor between that phase and an ideal gas, can be derived from:

$$\beta = \prod_{i}^{3N} \frac{u_{ih}}{u_{il}} \frac{e^{-\frac{1}{2}u_{ih}}}{1-e^{-u_{ih}}} \frac{1-e^{-u_{il}}}{e^{-\frac{1}{2}u_{il}}} \quad (1)$$

where $h$ and $l$ represent the heavy and light isotopes, respectively, and $N$ is the number of atoms in the unit cell. $u_{ih}$ and $u_{il}$ are defined as $u_{ih\ or\ il} = \hbar \omega_{ih\ or\ il}/k_B T$, where $\hbar$ and $k_B$ are the Planck and Boltzmann constants, respectively; $T$ is the temperature in Kelvin; and $\omega_{ih\ or\ il}$ is the vibrational frequency. Under the high-temperature approximation, the Bigeleisen–Mayer equation can be written as:

$$\beta = 1 + \left(\frac{1}{m} - \frac{1}{m'}\right) \frac{\hbar^2}{8k_B^2 T^2} <F> \quad (2)$$

where $m$ and $m'$ refer to the light and heavy isotopes, respectively, and $<F>$ is the force constant. The derivation process of Eq. (2) can be found in Wang et al.[36]. The equilibrium isotope fractionation factor ($10^3 \ln\alpha$) between the two phases is:

$$10^3 \ln\alpha_{A-B} = 10^3 \ln\beta_A - 10^3 \ln\beta_B = \left(\frac{1}{m} - \frac{1}{m'}\right) \frac{\hbar^2}{8k^2 T^2}(<F>_A - <F>_B) \quad (3)$$

This approach has been successfully applied to predict the equilibrium nickel and sulfur isotope fractionation between silicate and metallic melts[9,35]. The use of Eq. (3) requires the validity criterion that frequencies related to the element of interest $\omega_i$ (cm⁻¹) ≤ 1.39 T. For the temperature of core formation (>3000 K), the upper limit of frequencies is >4200 cm⁻¹, which is much higher than any vibrational frequency associated with N atoms.

The anharmonic effect becomes unignorable at high temperatures and it might significantly have a significant contribution to the $10^3 \ln\alpha$ based on the Bigeleisen–Mayer equation. Liu et al.[63] extensively discussed the anharmonic correction on the Bigeleisen–Mayer equation. Following the method in that work, we calculated the $10^3 \ln\beta$ of $N_2$ molecule with anharmonic correction. The relative difference in $10^3 \ln\beta$ between the results with and without anharmonic correction is only ~3%. This suggests that the anharmonic effect can only change the $10^3 \ln\alpha$ of $^{15}N/^{14}N$ by ~6%, consistent with the estimate in previous studies[63,64].

## First-principles molecular dynamics simulations

To calculate the force constants of N in silicate and metallic melts, we conducted FPMD simulations on silicate and metallic melts based on density functional theory (DFT) using VASP with the projector-augmented wave (PAW) method[65]. The generalized-gradient approximation (GGA)[66] was adopted for the exchange-correlation functional and the PBE pseudopotentials were used. The energy cutoff for the plane wave was 600 eV and the gamma point was used for the Brillouin zone summations over the electronic states. All FPMD simulations were conducted in the NVT thermodynamic ensemble with a fixed temperature of 3000 K controlled by a Nosé thermostat. The time step is 1 fs and the total simulation time is ~60 ps. The initial liquid structures were prepared by melting the configurations at 6000 K for 20 ps. Pressures at different volumes can be derived by averaging the pressure for each time step after equilibration (about 10 ps).

We used two different compositions, $Mg_{32}Si_{32}O_{96}NH_3$ and $Mg_{32}Si_{32}O_{95}N_2$, to model silicate melts under relatively reducing and oxidizing conditions, respectively. The chemical composition of $MgSiO_3$ was chosen for silicate melts because it has MgO and $SiO_2$ contents similar to those of primitive chondrites. We also performed FPMD simulations on N-bearing pyrolite, $Mg_{30}NaCa_2Fe_4Si_{24}Al_3O_{89}NH_3$ (pyrolite+$NH_3$) and $Mg_{30}NaCa_2Fe_4Si_{24}Al_3O_{89}N_2$ (pyrolite+$N_2$), to check the effect of other components on the structural properties of N. For metallic melts, we first focused on a simple system with a formula of $Fe_{98}N_2$ at different pressures and then checked the effect of other possible light elements in the core using a multicomponent alloy ($Fe_{87}Ni_4Si_6S_2C_2OH_5N_2$). We did not introduce a Hubbard U correction for Fe atoms in our calculations because Caracas et al.[67] checked the behavior of the Fe-bearing melt based on DFT + U and found that a +U correction does not significantly change the calculated results. The cell parameters and volumes of the simulated boxes are listed in Supplementary Table 1.

We extracted a large number of snapshots from the FPMD trajectories every 250 steps after equilibration and performed optimization solely on the N atomic positions for each snapshot. This involved relaxing the N atomic positions freely while fixing the positions of other atoms except for N. The relaxation of single atoms does not alter the positions of other atoms around N atoms, thereby preserving the structural information from the initial snapshot to the fullest extent, although the N atomic positions may undergo slight changes compared to the initial configurations. Subsequently, we applied seven different small displacements to the N atomic positions along each direction and computed the static energies of these configurations. The force constant is obtained as the second derivative of energy with respect to displacement. Consequently, the force constant matrix of N in each snapshot can be determined by fitting the relationship between static energies and small displacements with a second-order polynomial. The statistical average across all snapshots yields the average force constant of N in the melts (Supplementary Figs. 5–8). The errors encompass the error arising from the second-order polynomial fitting and the statistical error (±1σ deviation).

We extracted a large number of snapshots from the FPMD trajectories every 250 steps after equilibration and conducted optimization only on N atomic positions for each snapshot. This involved relaxing the N atomic positions freely while fixing the positions of other atoms except for N. The single-atom relaxation does not change the positions of other atoms around N atoms; hence, the structural information in the initial snapshot is maximally preserved, although the N atomic positions are slightly changed compared to the initial configurations. We then applied seven different small displacements to the N atomic positions along each direction (X, Y, and Z) and calculated the static energies of these configurations. The force constant is obtained as the second derivative of energy with respect to displacement. Thus, the force constant matrix of N in each snapshot can be calculated by fitting the relationship between static energies and small displacements with a

second-order polynomial[9]. The statistical mean of all snapshots yields the average force constant of N in the melts (Supplementary Figs. 5–8). The errors encompass the uncertainty arising from the second-order polynomial fitting and the statistical error (±1σ deviation)[9].

## Structural properties of nitrogen in silicate and metallic melts

To obtain the structural properties of N-bearing silicate and metallic melts, the radial distribution function (RDF) between two species A and B was calculated from:

$$g_{A-B}(r) = \frac{N}{\rho N_A N_B}\left\langle \sum_{i=1}^{N_A}\sum_{j=1}^{N_B}\delta\left(\vec{r} - \vec{R}_i^A + \vec{R}_j^B\right)\right\rangle \qquad (4)$$

where A and B refer to two species of interest; $N$ is the total number of atoms; $\rho$ is the atomic number density; and $N_A$ and $N_B$ refer to the total number of species A and B atoms, respectively. $\vec{R}$ represents the coordinates of these atoms. The coordination number (CN), which represents the number of B atoms distributed around A atoms, can be derived from the RDF.

Our calculations show that the N-N distance in the $Mg_{32}Si_{32}O_{95}N_2$ melt is ~1.12 Å at 5.6–99.6 GPa (Supplementary Fig. 1), which is similar to the N–N bonding length in nitrogen molecules ($N_2$). The N-Si and N-Mg distances are much longer than the N-N distance. When the cutoff for the coordination shell is 1.4 Å, the CN for the N-N pair is one, and no N-Si or N-Mg pairs form within this distance (Supplementary Fig. 1), suggesting that the N-N bond forms as a nitrogen molecule in the $Mg_{32}Si_{32}O_{95}N_2$ melt. Under relatively reducing conditions as modeled by the $Mg_{32}Si_{32}O_{96}NH_3$ melt, the N-Si, N-Mg, and N-H distances are ~1.72, ~2.02, and ~1.04 Å (Supplementary Fig. 2), respectively. When the cutoff for the coordination shell is 2.1 Å, the CNs for the N-Si, N-Mg, and N-H pairs are ~2.0, ~0.5, and ~0.2 at 5.5 GPa (Supplementary Fig. 2), respectively, suggesting that N is mainly bonded to Si atoms in the $Mg_{32}Si_{32}O_{96}NH_3$ melt, with a small fraction of N-H bonds. The CN for the N-Si pair generally increases with pressure and is greater than 3 at 76.5–97.7 GPa (Supplementary Fig. 2).

In the $Mg_{30}NaCa_2Fe_4Si_{24}Al_3O_{89}N_2$ (pyrolite+$N_2$) melt, the N-N distance shows a strong peak at 1.15 Å at 5.3 GPa (Supplementary Fig. 3), which is slightly larger than that in the $Mg_{32}Si_{32}O_{95}N_2$ melt. The CN for the N-N pair is also one when the cutoff for the coordination shell is 1.4 Å (Supplementary Fig. 3), implying a strong N-N bond in the pyrolitic melt under relatively oxidizing conditions. In the $Mg_{30}NaCa_2Fe_4Si_{24}Al_3O_{89}NH_3$ (pyrolite+$NH_3$) melt, the N-Si distance shows a similar distribution to that in the $Mg_{32}Si_{32}O_{96}NH_3$ melt at 23.3 GPa, and the CN for the N-Si pair is also ~2.0 if the cutoff is 2.1 Å (Supplementary Fig. 3). Although the N-Fe, N-Mg, and N-H distances also show peak distributions at ~1.81, ~2.08, and ~1.02 Å, respectively, the corresponding CNs are only ~0.6, ~0.3, and <0.1 (Supplementary Fig. 3). This implies that N is still preferentially bonded to Si atoms in the pyrolitic melt under relatively reducing conditions, with a fraction of the N-Fe bond.

In the $Fe_{98}N_2$ melt, the N-Fe distance is ~1.84 Å, and the CN for the N-Fe pair increases from ~5 at 0.7 GPa to ~7 at 98.1 GPa when the cutoff for the coordination shell is 2.4 Å (Supplementary Fig. 4). In the $Fe_{87}Ni_4Si_6S_2C_2OH_5N_2$ melt, the N atom is also dominantly bonded to Fe atoms with an N-Fe distance of ~1.81 Å at 25.1 GPa (Supplementary Fig. 4), similar to the N-Fe distance in the $Fe_{98}N_2$ melt. The CN for the N-Fe pair is ~5.3 at 25.1 GPa when the cutoff is 2.4 Å, while those for other pairs are smaller than 0.2, suggesting that other light elements do not significantly change the bonding environment around N in the metallic melt. The N-Fe bond length in the metallic melt is longer than the N-Si and N-N bond lengths in the silicate melts under relatively reducing and oxidizing conditions, respectively.

Our results are generally consistent with the experimental findings[22,23,31–33] that N dissolves principally as $N_2$ in silicate melts under

relatively oxidizing conditions (the oxygen fugacity log $fO_2$ > IW-1.5; reported in log units relative to the iron-wüstite oxygen buffer, IW) but as $N^{3-}$ under relatively reducing conditions in the form of the N-H complex (IW-1.5 <log $fO_2$ < IW-3) and/or N-Si bonding (log $fO_2$ < IW-3). It should be noted that the log $fO_2$ values of the simulated silicate melts cannot be given by the FPMD calculations. The main difference between our calculations and experimental results is that the fraction of the N-H complex in the $Mg_{32}Si_{32}O_{96}NH_3$ melt is not high, with a CN of ~0.2 for the N-H pair, probably because there are not sufficient H atoms to form the $NH^{2-}$ complex. To check the effect of N and H concentrations on the formation of N-H bonds in silicate melts, we also performed FPMD simulations on the $Mg_{32}Si_{32}O_{96}N_3H_9$ melt. We find that the N-Si, N-Mg, and N-H distances are ~1.71, ~2.07, and ~1.03 Å at 6.2 GPa (Supplementary Fig. 5), respectively, and their CNs are ~1.8, ~0.3, and ~0.6. The CN for the N-H pair greatly increases compared with that in the $Mg_{32}Si_{32}O_{96}NH_3$ melt, suggesting the presence of a large fraction of N-H in the $Mg_{32}Si_{32}O_{96}N_3H_9$ melt.

## Force constants of nitrogen in melts

The force constants <F> of N in melt snapshots are shown in Supplementary Figs. 5–8, and the average values are plotted in Supplementary Fig. 9. The <F> in the $Mg_{32}Si_{32}O_{95}N_2$ melt is ~760 N/m and does not substantially change with pressure, while those in the $Mg_{32}Si_{32}O_{96}NH_3$ and $Fe_{98}N_2$ melt increase from 366.9 N/m at 5.5 GPa to 576.1 N/m at 97.7 GPa and from 168.8 N/m at 0.7 GPa to 287.4 N/m at 98.1 GPa, respectively. The <F> variations with pressure are mainly controlled by the structural properties: in the $Mg_{32}Si_{32}O_{95}N_2$ melt, N dominantly occurs as $N_2$, and the N-N bond length and the CN do not significantly change with pressure (Supplementary Fig. 1); in the $Mg_{32}Si_{32}O_{96}NH_3$ and $Fe_{98}N_2$ melts, N is mainly present in the form of N-Si and N-Fe, respectively, and their CNs substantially increase with pressure (Supplementary Figs. 2 and 4). Compared to the $Mg_{32}Si_{32}O_{95}N_2$ melt, the $Mg_{30}NaCa_2Fe_4Si_{24}Al_3O_{89}N_2$ melt has a slightly lower <F> value (698.7 N/m at 5.3 GPa) because it has a slightly longer N-N bond. The <F> of N in the $Mg_{30}NaCa_2Fe_4Si_{24}Al_3O_{89}NH_3$ melt is ~38 N/m lower than that in the $Mg_{32}Si_{32}O_{96}NH_3$ melt because the CN for N-Si in the $Mg_{30}NaCa_2Fe_4Si_{24}Al_3O_{89}NH_3$ melt is slightly smaller than that in the $Mg_{32}Si_{32}O_{96}NH_3$ melt at ~23.3 GPa (Supplementary Fig. 3). In contrast, due to the presence of a larger percentage of N-H in the $Mg_{32}Si_{32}O_{96}N_3H_9$ melt (Supplementary Fig. 5), the <F> is ~46 N/m larger than that in the $Mg_{32}Si_{32}O_{96}NH_3$ melt. In metallic melts, $Fe_{98}N_2$ and $Fe_{87}Ni_4Si_6S_2C_2OH_5N_2$ have similar <F> values, consistent with the similarity of N-Fe bonding between them (Supplementary Fig. 4).

The <F> difference between silicate and metallic melts is mainly affected by the N species in silicate melts, which depends on the log $fO_2$. Under relatively oxidizing conditions, the <F> difference between silicate and metallic melts decreases from ~570 N/m at ~5 GPa to ~480 N/m at ~98 GPa, while it increases from ~195 N/m at ~5 GPa to ~290 N/ma at ~98 GPa under relatively reducing conditions (Fig. S9). This indicates that the equilibrium N isotope fractionation ($10^3\ln\alpha$) between silicate and metallic melts increases with pressure under relatively reducing conditions but decreases with pressure under relatively oxidizing conditions. Regardless of the $fO_2$, the core would be preferentially enriched in $^{14}N$ relative to the silicate part during core formation.

## Nitrogen isotope fractionation during planetesimal evaporation and core formation

As a volatile element, N would have undergone significant vaporization during planetary accretion. Here, we use the residual N fraction ($f_{res}$) to describe the degree of N loss during evaporation; hence, the N concentration of a bulk planet ($C_{BP}$) after evaporation is $C_{BP}=C_{init}*f_{res}$, where $C_{init}$ refers to the initial N concentration of the building material. Following a Rayleigh distillation model, the N isotope composition of the bulk planet ($\delta^{15}N_{BP}$) after evaporation is given by:

$$\delta^{15}N_{BP} = \delta^{15}N_{init} + \Delta^{15}N_{vapor-melt}*\ln(f_{res}) \quad (5)$$

where $\delta^{15}N_{init}$ is the initial N isotope composition of building material and $\Delta^{15}N_{vapor-melt}$ is the net N isotope fractionation between vapor and melt. When planetesimals undergo evaporation in the presence of nebular $H_2$ under a total pressure of approximately $10^{-4}$ bar, previous numerical simulations[38] demonstrate that the net isotope fractionation will be equal to the equilibrium isotope fractionation between vapor and melt. Equilibrium isotope fractionation during planetesimal evaporation can explain the Mg and Si isotopic and elemental compositions of bulk Earth[38]. Such conditions for planetesimal evaporation were required to explain the observed S, Se, and Te isotope composition of the bulk silicate Earth (BSE)[9,39]. Thus, $\Delta^{15}N_{vapor-melt}$ will be equal to the equilibrium N isotope fractionation between the vapor phase and silicate melt ($10^3\ln\alpha_{vapor-silicate}$).

To calculate the $10^3\ln\alpha_{vapor-silicate}$, we conducted thermodynamic calculations using the GRAINS code[68] with solar abundances for the elements[40] to determine the N species in the vapor phase. This code calculates the minimum Gibbs free energy of a given system and outputs all the species when the system achieves chemical equilibrium. The solar abundances for the elements were used to calculate the equilibrium gas phases because the solar nebular would not have completely dissipated during planetesimal evaporation in the first several million years[69]. Planetesimals would undergo evaporation in the presence of nebular $H_2$ under a total pressure of $1e^{-4}$ bar. We also checked the effect of H concentration on the N species in the vapor phase by performing thermodynamic calculations with solar elemental abundances but with H concentration decreasing by one and four orders of magnitude, conditions that are more oxidizing than the solar nebular. The results show that regardless of the H concentration in the system, N in the vapor phase dominantly occurs as $N_2$ with a percentage of > 99.9 % (Supplementary Fig. 11). We further calculated the <F> of N in an $N_2$ molecule using first-principles calculations. The atomic positions of an $N_2$ molecule in a cubic box (20 Å × 20 Å × 20 Å) were relaxed, and subsequently, <F> was derived using the small displacement method (Supplementary Table 1).

During core-mantle differentiation, equilibrium between the core and mantle is given by the N partition coefficient between metallic and silicate melts[3]:

$$D_N^{metal/silicate} = C_{core}/C_{BS} \quad (6)$$

where $C_{BS}$ and $C_{core}$ represent the N concentrations in the bulk silicate part and core, respectively. Based on the mass balance, the total mass of N in a bulk planet is conserved in these two reservoirs:

$$C_{BP} = M_{BS}*C_{BS} + (1 - M_{BS})*C_{core} \quad (7)$$

where $M_{BS}$ is the mass fraction of the bulk silicate part. Following the Rayleigh distillation model, the N isotope composition of the bulk silicate part ($\delta^{15}N_{BS}$) is given by:

$$\delta^{15}N_{BS} = \delta^{15}N_{BP} - 10^3\ln\alpha_{silicate-metal}*\ln(f_{BS}) \quad (8)$$

where $10^3\ln\alpha_{silicate-metal}$ is the equilibrium N isotope fractionation between silicate and metallic melts and $f_{BS}$ is the N fraction remaining in the bulk silicate reservoirs. If adopting the equilibrium model, then the $\delta^{15}N_{BS}$ will be:

$$\delta^{15}N_{BS} = \delta^{15}N_{BP} + (1 - f_{BS})*10^3\ln\alpha_{silicate-metal} \quad (9)$$

The choice of the core-forming model might affect the estimate of the effect of core-mantle differentiation on the $\delta^{15}N_{BS}$ (Supplementary

Fig. 10). Here we use two endmember models (the Rayleigh distillation model vs. the equilibrium model) to estimate the effect of core formation. The Rayleigh distillation model shows that core-mantle differentiation can only shift the BSE's $\delta^{15}N$ by at most +2‰ and +5‰ under relatively reducing and oxidizing conditions (Fig. 1), respectively. Compared to the Rayleigh distillation model, the variation in $\delta^{15}N_{BS}$ is smaller, at most +1.2‰, based on the equilibrium model (Supplementary Fig. 10). Consequently, the choice of core-forming model does not affect our conclusions that core formation cannot explain the $\delta^{15}N$ difference between enstatite chondrites and the bulk silicate Earth. We also modeled the combined effect of protoplanetary differentiation on the $\delta^{15}N$ of planetesimals using both the Rayleigh distillation and equilibrium models for core formation, and the results show very small differences under different redox conditions (Fig. 2 and Supplementary Fig. 12).

Earth had a protracted growth history with a change in the composition of accreting material from reducing to oxidizing, although the trajectory of its accretionary path is debated. The corresponding log $fO_2$ of core-mantle differentiation of Earth ranges from ~IW-3 to ~IW-1 (refs. 42,43). The $fO_2$ significantly affects the $D^{metal/silicate}_N$ and the N species in silicate melts, which further strongly impacts the N isotope fractionation caused by planetesimal evaporation and core formation. Therefore, in addition to using a fixed log $fO_2$ for Earth's accretion and differentiation, we also modeled the N concentration and $\delta^{15}N$ of the bulk silicate Earth by considering the dependences of $D^{metal/silicate}_N$, $10^3 ln\alpha_{vapor-silicate}$, and $10^3 ln\alpha_{silicate-metal}$ on the log $fO_2$. $D^{metal/silicate}_N$ under graphite-undersaturated conditions were reported by Grewal et al.[3] as a function of log $fO_2$. At log $fO_2$ > IW-1.5, N dissolves principally as $N_2$ in silicate melts, and the <F> of N in $Mg_{32}Si_{32}O_{95}N_2$ melt was used to calculate $10^3 ln\alpha_{vapor-silicate}$ and $10^3 ln\alpha_{silicate-metal}$. At log $fO_2$ < IW-1.5, N mainly occurs as $N^{3-}$, and the <F> of N in the $Mg_{32}Si_{32}O_{95}NH_3$ melt was used to determine these N isotope fractionation factors. Assuming that Earth accreted with a uniformly evolving oxygen fugacity from ~IW-3 to ~IW-1, that is, an even compositional distribution of accreting materials at log $fO_2$ values of ~IW-3 to ~IW-1, we estimated the $\delta^{15}N$ of the BSE with different initial values for building materials as well as the N concentration based on our results and revealed the accretion pattern of Earth's volatile elements.

## Data availability
The data that support the findings of this study is available in the article and Supplementary Information files.

## Code availability
The Vienna Ab Initio Simulation Package is proprietary software available for purchase at https://www.vasp.at/.

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

## Acknowledgements

This study is supported by the National Natural Science Foundation of China (42373008 and 41925017), the Fundamental Research Funds for the Central Universities (WK2080000189), and a Carnegie Science Fellowship to W.W. S.H. acknowledges support from NSF (AST-1910955 and EAR-2244895). Calculations were conducted at the Supercomputing Center of the University of Science and Technology of China. We acknowledge Wenshuai Zhang for their help in High-Performance Computing.

## Author contributions

W.W. conceived and designed this project. W.W. performed first-principles calculations and did the GRAINS calculations. W.W. wrote the manuscript, and M.J.W., J.P.B., and S.H. contributed to the interpretation of the results and revision of the manuscript.

## Competing interests

The authors declare no competing interests.
