## [Peer Review File · Nature Communications]

REVIEWER COMMENTS

Reviewer #1 (Remarks to the Author):

Reviews attached

Reviewer #2 (Remarks to the Author):

The manuscript describes new first principles modelling of processes fractionating nitrogen isotopes during Earth's accretion and differentiation. The aim was to test the hypothesis of accretion from enstatite chondrite and carbonaceous chondrite sources, plus losses via escape, and answer the question of the importance of a late veneer to the volatiles important to life.

The methodology is reasonable and has successfully been applied in other similar studies using first principles modelling of partition and diffusion. The modelling appears valid and it's a useful addition to the debate about the origin of essential volatiles. The authors conclude that nitrogen isotopes rule out a solely enstatite chondrite source for the Earth's volatiles and indicate that a late veneer of 30 - 100% of nitrogen is required to form the bulk silicate Earth.

The conclusions of the authors study corroborates measurements and modelling of other volatiles such as oxygen, sulfur and carbon, adding to the sum of knowledge about the importance of late accretion. However, the study shows that the proportions of late veneer predicted by vary between volatile elements and models.

The manuscript considers isotopes of a single element but would be stronger if it integrated the isotope variations between several volatile elements to develop a holistic model for life-essential volatiles. Other studies, notably of noble gases, consider multiple elements and isotopes to develop a more holistic, although still no conclusive, solution to the debate.

The study added incrementally to the debate about a later veneer but doesn't fundamentally change the debate.

Minor comments on the manuscript

Lines 41-42: It is not necessary to define δ here. If necessary refer to a text book.

Lines 52-53: The phrase 'unfortunately the necessary data are currently poorly known' does not support the work and could usefully be removed.

Line 55: I don't understand the wording 'only several'

Line 186: reword. The amount of late veneer required to explain the observed terrestrial volatiles depends on how much N is ...

Review of Wang et al. (2024)

“Nitrogen isotope constraints on Earth's late veneer”

Two competitive hypotheses about the inventory and accretion of life-essential volatile elements such as H, N, C, and S are (a) Earth might have accreted from volatile-rich materials and (b) Late veneer hypothesis where Earth was free of volatile when it accreted, and volatile is added by a late veneer of chondrite-like volatile-rich materials. In this manuscript, authors provide insight in the origin of these volatile on Earth by utilizing change in nitrogen partitioning in bulk silicate reservoir caused by core-mantle differentiation. In the manuscript, author use *first-principles* molecular dynamics simulation (FPMD) to study how nitrogen isotopes are fractionated during accretion and subsequent differentiation. The calculation presented in the manuscript is robust and the interpretation about the fractionation of N isotopes for various scenario presented here is supported by data. The manuscript is written well and below are some important findings of the manuscript.

1. Both the early differentiation and late veneer is responsible for present-day bulk silicate Earth's N if enstatite-chondrite-like Earth model is considered and 30-100% was contributed by late addition of carbonaceous-chondrite-like material.
2. The implication of small mass of late veneer are relatively small amounts of volatile elements and highly siderophile element (HSE) supply are from late veneer.

I think Nature Communication is a good venue for this manuscript. Focusing on the FPMD calculations which is expertise, I don't see any problem with the relaxation of structure, calculation of bond lengths and coordination numbers. However, I would suggest author to explain the details of estimating force constant of N. The graph presented in Figs. S5-S8 are nice but the description of how to get force constant which is explained in method sections (lines #462-471) is somewhat incomplete. For example, what is the procedure to optimize only on N positions from snapshots extracted every 250 steps? Another step in the calculation of force constant is applying 7 different small displacements to the N atomic positions to calculate static energy. Why seven? Has this method been established in the literature, or this is developed by the authors? As there are no references in the paragraph and without more details it is unclear. Since the equilibrium isotope fractionation factor is calculated using the force constant and this manuscript is main conclusion is based on nitrogen isotope fractionation during planetesimal evaporation and core formation, I think it would be useful to fully describe how force constant and how uncertainties on force constant are estimated.

Reviewer #1

Comment 1

Two competitive hypotheses about the inventory and accretion of life-essential volatile elements such as H, N, C, and S are (a) Earth might have accreted from volatile-rich materials and (b) Late veneer hypothesis where Earth was free of volatile when it accreted, and volatile is added by a late veneer of chondrite-like volatile-rich materials. In this manuscript, authors provide insight in the origin of these volatile on Earth by utilizing change in nitrogen partitioning in bulk silicate reservoir caused by core-mantle differentiation. In the manuscript, author use *first-principles* molecular dynamics simulation (FPMD) to study how nitrogen isotopes are fractionated during accretion and subsequent differentiation. The calculation presented in the manuscript is robust and the interpretation about the fractionation of N isotopes for various scenario presented here is supported by data. The manuscript is written well and below are some important findings of the manuscript.

1. Both the early differentiation and late veneer is responsible for present-day bulk silicate Earth's N if enstatite-chondrite-like Earth model is considered and 30-100% was contributed by late addition of carbonaceous-chondrite-like material.
2. The implication of small mass of late veneer are relatively small amounts of volatile elements and highly siderophile element (HSE) supply are from late veneer.

I think Nature Communication is a good venue for this manuscript. Focusing on the FPMD calculations which is expertise, I don't see any problem with the relaxation of structure, calculation of bond lengths and coordination numbers.

Reply: Thanks for the summary and publication recommendation.

Comment 2

However, I would suggest author to explain the details of estimating force constant of N. The graph presented in Figs. S5-S8 are nice but the description of how to get force constant which is explained in method sections (lines #462-471) is somewhat incomplete. For example, what is the procedure to optimize only on N positions from snapshots extracted every 250 steps? Another step in the calculation of force constant is applying 7 different small displacements to the N atomic positions to calculate static energy. Why seven? Has this method been established in the literature, or this is developed by the authors? As there are no references in the paragraph and without more

details it is unclear. Since the equilibrium isotope fractionation factor is calculated using the force constant and this manuscript's main conclusion is based on nitrogen isotope fractionation during planetesimal evaporation and core formation, I think it would be useful to fully describe how the force constant and how uncertainties on the force constant are estimated.

Reply: Thanks for the suggestions. The procedure for optimizing only N positions from snapshots involves two steps. First, we fix the positions of atoms except for N, and then we conduct structural relaxation, allowing the N atomic positions to relax freely. Subsequently, we obtain the force constant of N in each snapshot using the small-displacements method. This entails applying seven different small displacements to the N atomic positions along each direction (X, Y, and Z) and calculating the static energies of these configurations. The relationship between static energies and displacements follows a second-order polynomial, with the force constant being the second derivative of energy with respect to displacement. Seven small displacements suffice to establish a high-level second-order polynomial relationship between energy and displacement (see the figure below). This method has also been used to calculate the force constant of S in our previous studies (Wang et al., 2021).

We added more details to clarify the relaxation and calculations of the force constant in the Method part:

“We extracted a large number of snapshots from the FPMD trajectories every 250 steps after equilibration and conducted optimization only on N atomic positions for each snapshot. This involved relaxing the N atomic positions freely while fixing the positions of other atoms except for N. The single-atom relaxation does not change the

positions of other atoms around N atoms; hence, the structural information in the initial snapshot is maximally preserved, although the N atomic positions are slightly changed compared to the initial configurations. We then applied seven different small displacements to the N atomic positions along each direction (X, Y, and Z) and calculated the static energies of these configurations. The force constant is obtained as the second derivative of energy with respect to displacement. Thus, the force constant matrix of N in each snapshot can be calculated by fitting the relationship between static energies and small displacements with a second-order polynomial. The statistical mean of all snapshots yields the average force constant of N in the melts (Figs. S5-S8). The errors encompass the uncertainty arising from the second-order polynomial fitting and the statistical error ($\pm 1\sigma$ deviation).”

Reviewer #2

Comment 1

The manuscript describes new first principles modelling of processes fractionating nitrogen isotopes during Earth’s accretion and differentiation. The aim was to test the hypothesis of accretion from enstatite chondrite and carbonaceous chondrite sources, plus losses via escape, and answer the question of the importance of a late veneer to the volatiles important to life.

The methodology is reasonable and has successfully been applied in other similar studies using first principles modelling of partition and diffusion. The modelling appears valid and it’s a useful addition to the debate about the origin of essential volatiles. The authors conclude that nitrogen isotopes rule out a solely enstatite chondrite source for the Earth’s volatiles and indicate that a late veneer of 30 - 100% of nitrogen is required to form the bulk silicate Earth.

The conclusions of the authors study corroborates measurements and modelling of other volatiles such as oxygen, sulfur and carbon, adding to the sum of knowledge about the importance of late accretion. However, the study shows that the proportions of late veneer predicted by vary between volatile elements and models.

Reply: Thanks for the summary.

Comment 2

The manuscript considers isotopes of a single element but would be stronger if it integrated the isotope variations between several volatile elements to develop a holistic

model for life-essential volatiles. Other studies, notably of noble gases, consider multiple elements and isotopes to develop a more holistic, although still no conclusive, solution to the debate. The study added incrementally to the debate about a later veneer but doesn't fundamentally change the debate.

Reply: Thanks for the suggestion. We do agree with the reviewer that noble gases may provide important constraints on the origin of Earth's volatiles, but they overall suggest a more complex story. For instance, the high Ne isotope ratio observed in the primordial plume mantle indicates the preservation of nebular gases in the deep mantle (Williams and Mukhopadhyay, 2019). However, the isotopic compositions of mantle Kr and Xe align with chondritic values, yet atmospheric Kr and Xe display non-chondritic signatures (Holland et al., 2009). Notably, compared to the atmosphere, the mantle exhibits depletion in the lighter isotopes of Kr but enrichment in the lighter isotopes of Xe (Holland et al., 2009). This discrepancy suggests that atmospheric Kr and Xe isotopic compositions cannot solely be accounted for by mantle outgassing or hydrodynamic escape. The atmosphere harbors over 90% of the budget of Kr and Xe in the bulk silicate Earth (BSE) (Halliday, 2013; Marty, 2012). The disparities in isotopic compositions and budgets of Kr and Xe between the atmosphere and the mantle imply that the contemporary Kr and Xe in the atmosphere likely originated from a late veneer (Mukhopadhyay and Parai, 2019).

However, the proposition of late delivery of noble gases via carbonaceous-chondrite-like material encounters challenges. First, the isotopic composition of Xe in the atmosphere does not align with a known chondritic source (Pepin, 1991). Second, if late accreting material possessed a CI chondrite composition, it would need to constitute approximately 4.5% of Earth's mass to account for the Kr budget (Marty, 2012), which would lead to overabundances in the BSE's budget of many volatiles as well as platinum group elements. It was late argued that accretion of a small amount of cometary ice may explain atmospheric Xe, but it leads to no detectable change in the H, or C budget of Earth (Halliday, 2013; Marty et al., 2016, 2017).

Various noble gases convey distinct narratives regarding the origin of Earth's volatiles. This divergence could be linked to the enigmatic reservoirs of noble gas in the deep mantle and the noble gas isotope fractionation occurring during planetary differentiation. We are currently endeavoring to narrow down the noble gas isotope fractionation resulting from planetary differentiation, although this will be the focus of a separate paper.

We now added more discussion regarding the noble gases in lines 223-230:

“Additionally, noble gases provide significant insights into the origin of Earth’s volatiles, albeit presenting a more intricate narrative of their origin and evolution⁵⁵⁻⁵⁹. For instance, the high Ne isotope ratio observed in the primordial plume mantle indicates the preservation of nebular gases in the deep mantle⁵⁹, while the isotopic composition of heavy Kr and Xe, primarily residing in the atmosphere, has been attributed to a late delivery of carbonaceous-chondrite-like material⁶⁰ or cometary ice⁶¹. Further high-precision investigation into mantle reservoirs of noble gases, as well as noble gas isotope fractionation during planetary differentiation⁶², holds promise in understanding the origin of Earth’s noble gases.”

Comment 3

Lines 41-42: It is not necessary to define del here. If necessary refer to a text book.

Reply: Done.

Comment 4

Lines 52-53: The phrase ‘unfortunately the necessary data are currently poorly known’ does not support the work and could usefully be removed.

Reply: We rewrote this sentence.

Comment 5

Line 55: I don’t understand the wording ‘ only several’

Reply: We rephrased this sentence.

Comment 6

Line 186: reword. The amount of late veneer required to explain the observed terrestrial volatiles depends on how much N is ...

Reply: We rewrote this sentence.

REVIEWERS' COMMENTS

Reviewer #1 (Remarks to the Author):

I am happy with the changes authors have made to the method section to explain the calculation of force constant. I do not have any further comments.